# Donanemab for Alzheimer’s Disease: A Systematic Review of Clinical Trials

**DOI:** 10.3390/healthcare11010032

**Published:** 2022-12-22

**Authors:** Areeba Rashad, Atta Rasool, Muhammad Shaheryar, Azza Sarfraz, Zouina Sarfraz, Karla Robles-Velasco, Ivan Cherrez-Ojeda

**Affiliations:** 1Department of Research and Publications, Fatima Jinnah Medical University, Lahore 54000, Pakistan; 2Department of Research, Services Institute of Medical Sciences, Lahore 54000, Pakistan; 3Department of Research, Rawal Institute of Health Sciences, Islamabad 45550, Pakistan; 4Department of Pediatrics and Child Health, The Aga Khan University, Karachi 74800, Pakistan; 5Department of Allergy, Immunology & Pulmonary Medicine, Universidad Espíritu Santo, Samborondón 092301, Ecuador

**Keywords:** Alzheimer’s disease, donanemab, plaque, cognition, elderly care

## Abstract

Amyloid-β (Aβ) plaques and aggregated tau are two core mechanisms that contribute to the clinical deterioration of Alzheimer’s disease (AD). Recently, targeted-Aβ plaque reduction immunotherapies have been explored for their efficacy and safety as AD treatment. This systematic review critically reviews the latest evidence of Donanemab, a humanized antibody that targets the reduction in Aβ plaques, in AD patients. Comprehensive systematic search was conducted across PubMed/MEDLINE, CINAHL Plus, Web of Science, Cochrane, and Scopus. This study adhered to PRISMA Statement 2020 guidelines. Adult patients with Alzheimer’s disease being intervened with Donanemab compared to placebo or standard of care in the clinical trial setting were included. A total of 396 patients across four studies received either Donanemab or a placebo (228 and 168 participants, respectively). The Aβ-plaque reduction was found to be dependent upon baseline levels, such that lower baseline levels had complete amyloid clearance (<24.1 Centiloids). There was a slowing of overall tau levels accumulation as well as relatively reduced functional and cognitive decline noted on the Integrated Alzheimer’s Disease Rating Scale by 32% in the Donanemab arm. The safety of Donanemab was established with key adverse events related to Amyloid-Related Imaging Abnormalities (ARIA), ranging between 26.1 and 30.5% across the trials. There is preliminary support for delayed cognitive and functional decline with Donanemab among patients with mild-to-moderate AD. It remains unclear whether Donenameb extends therapeutic benefits that can modify and improve the clinical status of AD patients. Further trials can explore the interplay between Aβ-plaque reduction and toxic tau levels to derive meaningful clinical benefits in AD patients suffering from cognitive impairment.

## 1. Introduction

Alzheimer’s disease (AD) is the leading cause of dementia and has been recognized as a global public health priority [1]. AD was present in nearly 5.8 million people in the United States in 2020 and this number is projected to triple to 14 million people in 2060 [2]; the global prevalence is predicted to be 152.8 million cases in 2050 [3]. Numerous clinical trials have failed for effective disease-modifying treatment in AD over the past 20 years [4,5,6,7,8]. Removal of the extracellular amyloid-β (Aβ) plaques in the brain is one of the mechanisms explored for treating AD. Molecularly, AD has two pathological hallmarks in the central nervous system (CNS), extracellular amyloid plaques derived from amyloid-β peptides and the formation of neurofibrillary tangles consisting of aggregated and hyperphosphorylated tau [9,10]. Various clinical trials utilizing therapeutic compounds tend to target a reduction in amyloid plaques; these have shown notable correlations in amyloid-related imaging abnormalities (ARIA) rates, plaque removals, and efficacy of treatment [11,12]. Anti-amyloid immunotherapy, Aducanumab, was given conditional approval by the Food and Drug Administration (FDA), providing support for the amyloid hypothesis as a valid approach for targeting Aβ plaques [13]. 

Recently, anti-amyloid immunotherapy, Donanemab, is one of few therapies that is emerging as a promising candidate to significantly reduce cerebral amyloid deposits [12,14]. Donanemab is a humanized antibody and acts against the N-truncated pyroglutamate amyloid-β peptide at position 3 (pGlu3-Aβ, AβpE3) and is currently being investigated as a treatment for AD [15]. Approaches to target pGlu3-Aβ have so far included the reduction in pGlu3-Aβ formation at the glutamyl cyclase (QC) catalyzing of N-truncated Aβ to form pGlu-Aβ and anti-pGlu3-Aβ antibodies. Antibodies including Donanemab aim to clear pGlu-Aβ after formation and/or blocking aggregation [16,17,18,19,20]. AβpE3 antibodies have different binding properties against either soluble or aggregated conformations of AβpE3-42 [15]. Donenemab has been shown to have strong action with amyloid plaques, specifically cored plaques in the CNS [21]. There is, however, a lack of clarity regarding its beneficial effects for AD therapy. 

We aimed to systematically collate all clinical trials of Donanemab administered in AD patients. In this systematic review, we summarized the design and inclusion criteria, dosing regimens, primary and secondary outcome measures, the efficacy of treatment, and safety measures.

## 2. Materials and Methods

A comprehensive systematic search was conducted across the following databases adhering to PRISMA Statement 2020 guidelines: PubMed/MEDLINE, CINAHL Plus, Web of Science, Cochrane, and Scopus. No language or date of inception restrictions was applied. The search terms across the databases comprised a combination of the following: LY3002813, Donanemab, and Alzheimer’s Disease. The Boolean (And/Or) logic was applied. The titles and abstracts of all the studies from the enlisted databases were screened together by two reviewers (A.S. and Z.S.). During the screening phase, the reference lists were additionally reviewed. The studies were searched through November 5, 2022. A third reviewer was present to resolve any discrepancies (I.C.-O.). Cohen’s coefficient of the inter-reviewer agreement was computed using SPSS v.25. 

Only clinical trials either randomized or non-randomized were eligible for inclusion in this review. All other studies including cohorts (prospective or retrospective), case series/reports, systematic reviews and meta-analytical studies, brief reports, and letters to editors were excluded.

Adult patients with Alzheimer’s disease being intervened with Donanemab compared to placebo or standard of care in the clinical trial setting were included. The outcomes comprised of measures of brain amyloid plaque levels and tests, including the ADAS-Cog13 (the 13-item cognitive subscale of the Alzheimer’s Disease Assessment Scale), ADAS-Cog14 (the 14-item cognitive subscale of the Alzheimer’s Disease Assessment Scale), ADCS-iADL (the Alzheimer’s Disease Cooperative Study–Instrumental Activities of Daily Living Inventory), ADCS-MCI-ADL-24 (Alzheimer’s Disease Cooperative Study–Activities of Daily Living–Mild Cognitive Impairment 24-item version), CDR-SB: (Clinical Dementia Rating Scale–Sum of Boxes), MMSE (Mini-Mental State Examination), and the NTB (Neuropsychological Test Battery).

All reviewers extracted data together from the studies into a shared spreadsheet. An identification of the following components was made and tabulated: serial number, author, year, title, journal, phase, design, inclusion criteria, pharmacologic intervention, outcome measures, follow-up, sample size, age (years), gender, the severity of Alzheimer’s disease, APOE-ε4 carriers, efficacy and safety of the intervention. The individual study data were entered in a presentable format during the extraction phase. The software Endnote X9 (Clarivate, London, UK) was used for omitting duplicates during the study selection process. The bibliographic referencing software used in this study was Mendeley (Elsevier, Amsterdam, The Netherlands). 

Version 2 of the Cochrane risk-of-bias tool for randomized trials (RoB 2) was used to assess the risk of bias (quality) in the included trials. The tool comprised five domains: first, any biases presented during the randomization process were assessed. Second, any risks arising due to deviations from intended interventions were noted. Third, biases arising at any point due to missing outcome data were reviewed. Fourth, biases in the measurement of the outcomes were assessed in the included trials. Fifth, biases presented in the selection of the reported result were assessed as well. As a result, domain-level judgments were made and classified as follows: (1) low risk, (2) some concerns, and (3) high risk. A weighted summary plot was illustrated depicting the risk of bias assessment.

## 3. Results

The PRISMA flowchart is attached in Figure 1. The Kappa score was 0.934, suggesting excellent agreement between the reviewers.

A total of 396 patients were included in the trials. Of these, 228 patients (57.6%) received Donanemab and 168 patients (42.4%) were given a placebo. The gender ratio was balanced with 212 patients (53.5%) across all trials being female. Among the 2 studies that reported APOE-ε4 carrier status, 244 of 331 (73.7%) were carriers, of whom 56 patients (22.9%) were homozygous. The design, inclusion criteria, dosing regimen, outcome measures, and follow-up period are summarized in Table 1. The key patient characteristics, efficacy, and safety outcomes are elaborated on in Table 2. 

### 3.1. Design and Inclusion Criteria 

Lowe et al. [22] conducted an investigator- and subject-blinded, randomized, placebo-controlled study in phase I; the parallel-group study was followed by numerous doses and dose-escalation strategies [22]. Non-Japanese and Japanese participants originating from the United States and Japan were included. The conditions for inclusion were: non-fertile women or men aged 50 years or above presenting with memory impairment on the Free and Cued Selective Reminding Test Immediate Recall (FCSRT-IR; picture version), an ^18^F-flortaucipir positron emission tomography (PET) scan consistent with amyloid pathology, and a mini-mental state examination (MMSE) score between 16 and 30. Patients with mild cognitive impairment (MCI) due to Alzheimer’s disease (AD) or mild-to-moderate AD dementia were enrolled.

Lowe et al. [23] conducted a three-part, patient and investigator-blinded, randomized cohort, placebo-controlled, parallel-group, single- and multiple-dosage, phase IB study in the United States and Japan [23]. The inclusion criteria consisted of males and non-fertile females aged 50 or above with evidence of memory impairment on FCSRT-IR. Participants were required to have an MMSE score between 16 and 30, a clinical dementia rating (CDR) scale score between 0.5 and 2, a memory box score of 0.5 or more, and an ^18^F-flortaucipir PET scan consistent with amyloid pathology. Patients with mild cognitive impairment (MCI) due to AD or mild-to-moderate AD dementia were included. 

Mintun and colleagues [21] designed a phase II multi-center, randomized, double-blind, placebo-controlled trial called TRAILBLAZER-ALZ, which was conducted in the United States and Canada [21]. The inclusion criteria comprised patients between the age of 60 to 85 years with early symptomatic AD with gradual and progressive change in memory function for ≥6 months—this was defined as prodromal AD or mild AD with dementia, MMSE scores between 20 and 28, intermediate tau levels with ^18^F-flortaucipir PET tau SUVR between 1.10 and 1.46, and patients with topographically advanced AD despite an SUVR < 1.10, and elevated amyloid levels ≥ 37 CL. Shcherbinin and colleagues [24] reported post hoc findings of the multicenter, double-blind, phase II TRAILBLAZER-ALZ trial conducted by Mintun and colleagues [21]. The TRAILBLAZER-ALZ trial was conducted for up to 76 weeks and a 48-week follow-up period.

### 3.2. Dosing Regimens

Lowe and colleagues [22] established five cohorts to receive intravenous (IV) doses from 0.1 mg/kg to 10 mg/kg Donanemab intravenously against placebo and followed by a 12-week follow-up period for each dose level such that the lowest dose of 0.1 mg/kg Donanemab was sentinel. After establishing safety data at 7 days, the remaining participants were given higher doses at least 1 day apart. The multiple-ascending dose (MAD) phase was consequently given in five cohorts with AD patients for up to four doses of Donanemab once per month depending on the initial dose. Only cohort 1 ascended from 0.1 mg/kg to a higher dose of 0.3 mg/kg Donanemab during the MAD phase. Two arms were taken as a relative exposure assessment including an unblended, single, subcutaneous 3 mg/kg dose of Donanemab in AD patients and unblinded, single, intravenous 1 mg/kg dose of Donanemab in healthy patients.

Lowe et al. [23] established a total of six cohorts, of which cohorts 1–3 were given a single dosage of Donanemab each at 10 mg/kg, 20 mg/kg, and 40 mg/kg, respectively; cohort 4 was given multiple intravenous doses of Donanemab at 10 mg/kg every 2 weeks for 24 weeks. Lastly, cohorts 5–6 were given multiple intravenous doses of Donanemab at 10 mg/kg and 20 mg/kg every 4 weeks for 72 weeks. Overall, there were three dosing regimens: (i) a single dose of 10, 20, or 40 mg/kg, (ii) 10 mg/kg Q2weekly for 24 weeks, and (iii) 10 or 20 mg/kg Q4weekly for 16 months.

Mintun and colleagues [21] and Shcherbinin and colleagues [24] reported findings of the same trial which provided patients with 700 mg Q4weekly Donanemab for the first 3 months and then 1400 mg for up to 18 months.

### 3.3. Primary and Secondary Outcome Measures

Lowe et al. [22] focused on the safety and tolerability of single and multiple doses of Donanemab as the primary outcome. The secondary outcome was to assess the effect of donanemeb on brain plaque using an ^18^F-flortaucipir PET scan with SUVR values [22]. Other outcomes were cognitive improvement categorized by the Alzheimer’s Disease Assessment Scale Cognitive Subscale (ADAS-Cog-14), MMSE scores, and FCSRT-IR values obtained.

Lowe et al. assessed amyloid plaque load using an ^18^F-flortaucipir PET scan with SUVR values as the primary outcome [23]. Secondary outcomes included antidrug antibodies (ADAs) against Donanemab and blood pharmacokinetics of Donanemab. Other outcomes were cognition improvement as assessed by MMSE, FCSRT-IR, ADAS-Cog-14, the Alzheimer’s Disease Cooperative Study-Mild Cognitive Impairment-Activities of Daily Living, 24-item questionnaire (ADCS-MCI-ADL-24), and the Neuropsychological test battery (NTB) at baseline, 24, 48, and 72 weeks after initiation of treatment or discontinuation.

Mintun and colleagues quantified the change from baseline to the 76th week in the Integrated Alzheimer’s Disease Rating Scale (iADRS) scores, a combined cognitive/functional measure for early stage AD [21]. Secondary measures were changes from baseline in CDR, ADAS-Cog13, ADCS-iADL, and MMSE score outcomes, as well as amyloid and tau PET and volumetric magnetic resonance imaging (MRI).

Shcherbinin and colleagues [24] investigated the exploratory post hoc analyses of the TRAILBLAZER-ALZ trial. The primary outcome was the Donanemab-induced amyloid reduction, specifically the association between amyloid plaque clearance after 24 weeks and the effect on tau progression over 76 weeks. Three models were developed including (i) an exposure–response model of sustainability of amyloid reduction, (ii) a mediation model of reduction in amyloid plaque levels and tau pathology, and (iii) a disease-progression model of reduction in amyloid plaque levels and clinical benefits. 

### 3.4. Efficacy of Donanemab

Lowe et al. (2021) [22] found that Donanemab was well-tolerated up to 10 mg/kg dosages. One single-dose administration of that between 0.1 and 3 mg/kg led to a mean terminal elimination half-life of around 4 days, whereas this increased to 10 days with 10 mg/kg. Notably, at 24 weeks, the 10 mg/kg dose showed changes in amyloid PET scans with a 40–50% reduction in amyloid plaque levels with a mean SUVR change of −0.36 from 1.65, alongside a Centiloids (CL) change of −44.4 (SD: 14.2) from baseline. Anti-drug antibodies were developed in 90% of subjects 3 months after the single dose.

Lowe et al. [23] reported that Donanemab led to rapid amyloid reduction even after a single dosage. In the 24th week, the single-dose cohorts had an amyloid PET mean reduction of −16.5 CL with 10 mg/kg dosing, −40.0 CL with 20 mg/kg dosing, and −49.6 with 40 mg/kg dosing. In the multiple-dosage cohorts at week 24, the 10 mg/kg Q2weekly dosing arm had a −55.8 CL, the 10 mg/kg Q4weekly dosing arm had a −50.2 CL, and the 20 mg/kg Q4weekly dosing had a −58.4 CL reduction in mean amyloid levels. Two patients in the single-dose cohorts (one patient in the 20 mg/kg and one patient in the 40 mg/kg) and nine patients in the multiple-dose cohorts (two in the 10 mg/kg Q2weekly, two in the 10 mg/kg Q4weekly, and five in the 20 mg/kg Q4weekly) had a complete amyloid clearance status of below 24.2 CL. All but 1 of the 46 patients (97.8%) had positive treatment-emergent ADA (TE-ADA) with Donanemab.

Mintun and colleagues [21] reported significance in iADRS findings in the 76th week with a Donanemab-placebo difference of 32% (*p* = 0.04) in favor of Donanemab. Other findings were insignificant, including the fact that the difference between the Donanemab/placebo group in change from baseline at week 76 was −0.36 for CDR scores, −1.86 for ADAS-Cog13 scores, 1.21 for ADCS-iADL scores, 0.64 for the MMSE score. There was no significant change in global tau levels found on ^18^F-flortaucipir PET at week 76 when compared to baseline. At 52 and 76 weeks, there was a relatively greater decrease in whole-brain volume and a greater increase in ventricular volume in the Donanemab group compared to the placebo group. 

Shcherbinin and colleagues [24] reported that 46 (40%) of 115 participants receiving Donanemab achieved a full amyloid clearance threshold, computed at 24.1 CL, and this subset of patients had a lower baseline amyloid level (mean: 92.8 CL) compared to the entire group (mean: 107 CL) (r: −0.54 meaning moderate correlation between baseline and removed amyloid plaque level in the first 24 weeks). The amyloid clearance was sustained with a 0.02 (7.75) mean (SD) rate of re-accumulation over 1 year and those who achieved an amyloid level of ≤11 CL at week 24 and discontinued treatment would need a median period of 3.9 years [95% CI: 1.9–8.3 years] to reaccumulate amyloid to 24.1 CL. There was a 34% slowing of overall tau level accumulation categorized by neocortical SUVR in the entire Donanemab group compared to placebo at week 76 [24]. 

### 3.5. Safety and Tolerability of Donanemab

Lowe et al. [22] reported that 6 (16.2%) out of 37 participants with intravenous Donanemab dosing presented with mild or moderate infusion reactions (one patient—0.3 mg/kg arm, two patients—1.0 mg/kg arm, and three patients—3.0 mg/kg arm) post 2–3 doses. One patient discontinued due to infusion reactions; symptoms comprised of chills/shivering, fever, asymptomatic hypotension, flushing, myalgia, dyspnea, and rash. There were two patients that presented with asymptomatic amyloid-related imaging abnormalities due to cerebral microhemorrhages (ARIA-H), one patient in the 3mg/kg IV cohort and one patient in the 3 mg/kg SC single-dose cohort. Both ARIA-microhemorrhage adverse events occurred in patients who had baseline microhemorrhages. 

Lowe et al. [23] reported seven serious adverse events among six patients, of which one patient died due to non-drug-related myocardial infarction, one patient had intermittent symptomatic cerebral edema (ARIA-E), and the other four patients had non-drug-related events. Among the 46 participants that were intervened, 12 patients (26.1%) reported vasogenic edema events (ARIA-E) occurring in all but the 10 mg/kg single-dose cohort, 10 patients (21.7%) had cerebral microhemorrhage events (ARIA-H) across all intervention dosing cohorts, and 2 patients (4.4%) had superficial siderosis events (ARIA-H) in the 10 and 20 mg/kg Q4 weekly cohort. Two patients (4.4%) in the interventional arms in the 20 mg/kg Q4 weekly cohort discontinued Donanemab due to treatment-related adverse events (TRAE). 

Mintun and colleagues [21] also reported no noteworthy differences between the two groups in terms of mortality and serious adverse effects. Comparable to previous trials, 35 of 131 participants (26.7%) had an ARIA-E of whom 8 (6.1%) participants were symptomatic. The ARIA-H events were present in 40 of 131 (30.5%) participants of which microhemorrhage was noted in 26 (19.8%) participants and superficial siderosis in 23 (17.6%) participants in the Donanemab group. The infusion-related reaction was found in 10 of 131 (7.6%) participants in the intervention group. 

Shcherbinin and colleagues [24] reported that all-cause mortality was determined to be lower in Donanemab participants (0.76%) compared to placebo (1.6%). Overall, there were no differences in the serious adverse events when comparing the Donanemab and placebo groups (19.85% vs. 20%). 

### 3.6. Risk-of-Bias Synthesis

For the risk of bias arising during the randomization process, all studies had low concerns. On assessing the bias due to deviations from the intended interventions, two studies had some concerns, whereas two studies had low concerns. Bias due to missing outcome data was presented with some concerns in two studies and low concerns in two studies. A low risk of bias was noted due to the measurement of the outcome in all studies. Two studies each had some concerns and low concerns for bias in the selection of the reported result (Figure 2). 

## 4. Discussion

In this systematic review, we critically summarized evidence of all clinical trials to evaluate the efficacy and safety of Donanemab in AD patients. A total of four studies were found and a total of 396 patients were enrolled in the trials. Of these, 228 patients received Donanemab and 168 patients were given a placebo. The mean age ranged from 69.7 years to 75.2 years across the trials. Overall, the gender ratio was balanced with a total of 212 (53.5%) female patients. Dosing regimens were given based on weight or as a fixed dose, all given intravenously; 700–1400 mg or 10–40 mg/kg every 4 weeks for 72 weeks. One dose-escalation study by Lowe et al. (2021) established the 10 mg/kg dose as well-tolerated. The duration of follow-up varied from 48 to 76 weeks. The inclusion criteria were not comparable, including patients in different age groups, mild or moderate AD categorized by MMSE, and different criteria for severity of amyloid pathology on PET scan; while the mean age of the patients across the trials was comparable, the criteria for severity of AD was either mild or moderate. The tools used to assess primary efficacy outcomes included amyloid clearance threshold on Positron Emission Tomography (PET) scan with Standardized Uptake Value Ratio (SUVR), and the Integrated Alzheimer’s Disease Rating Scale (iADRS). The secondary outcomes included cognitive measures: CDR-SB scores, ADAS-Cog13 scores, and MMSE scores, as well as tau levels on PET-SUVR. Safety was established with the monitoring of Donanemab-related serious adverse events and mortality. Overall, the four trials did not have much overlap concerning inclusion criteria, dosing, and efficacy outcomes measures whereas age and gender were relatively uniform. 

There is a unique interplay between amyloid-β (Aβ) and tau which are both the main therapeutic targets for AD treatment currently being explored. The phase I/II trials included in this study targeted amyloid-β (Aβ) plaques which were reduced with Donanemab but depended on percent change. This meant that higher Aβ levels among the patients at baseline showed a prominent magnitude of reduction whereas lower Aβ levels of patients at baseline were found to have total amyloid clearance, defined as an amyloid plaque level of less than 24.1 CL, among 40% of participants receiving Donanemab (N = 131) in the phase II TRAILBLAZER-ALZ (NCT03367403) trial. Even single Donanemab doses reduced Aβ levels in the phase 1 trials. However, the lack of significant cognitive function improvement across the phase I/II trials can be underpinned by certain underlying mechanisms. The clinical status of AD patients correlates with both Aβ and tau and accumulating evidence suggests that soluble forms of both work together, independent of their accumulation in the central nervous system (CNS). Pathologically, in AD, Aβ is upstream of tau and triggers its conversion from a normal to a pathological state. However, there is evidence that tangled tau increases Aβ toxicity through bidirectional feedback. Exploring AD therapeutics requires a modest approach that would mean intervention before the accumulation of destructive Aβ plaques, tau tangles, and clinically present cognitive impairment. 

What is clear in the literature is that there is a long delay or lag between rising Aβ levels and tau deposition of an average of 13.3 years [25]. Evidence points towards an initial memory impairment due to amyloid-induced synaptic damage, tau accumulation, and neurofilament changes associated with later global cognitive impairment [26]. Explored in human samples, cognitive decline in AD is, however, primarily due to tau oligomers present in the synapses in synergy with Aβ oligomers [27]. In a similar light, abnormal levels of cortical tau and associated cognitive decline on PET are only found with Aβ levels above 40 Centiloids (CLs) [28]. This suggests a window between a rise in Aβ levels and the associated rise of toxic tau levels which must be considered a critical time to address underlying pathological sequelae of AD, specifically cognitive impairment, with Aβ-targeted reduction. It is likely that the cognitive improvement was not found across the Donanemab phase I/II trials, measured by MMSE, ADAS-Cog13, and CDR-SB scores, in this study as the pre-existing tauopathy cannot be completely confined in cognitively-impaired AD patients unless potentially in the absence of Aβ levels, e.g., complete amyloid clearance [29]. 

Regarding the lack of any measurable cognitive improvement, the concept of cognitive resilience as an underlying biological mechanism and as a therapeutic concept in the face of AD neuropathology [27]. Different levels of cognitive resilience at baseline among AD patients in trials are expected and can be explored as part of the treatment strategy which may be the key to improving cognition [30]. In longitudinal studies, the prodromal stage is expected to range from 1 year [31] to over 10 years [32] and is known to impact some cognitive domains more than others [33]. The prodromal phase known as mild cognitive impairment (MCI), a precursor to AD, is when accelerating cognitive decline occurs over many years [33]. It is reasonable to consider Aβ targeted therapies in MCI before the occurrence of global cognitive decline to derive maximum benefit which ties in with supporting cognitive resilience among patients at risk for AD. 

A point of contention in phase I/II Donanemab trials was the lack of change in global Tau levels. PET biomarkers and analytics have advanced with the Food and Drug Administration (FDA)-approved ^18^F radiotracers, particularly the ^18^F-flortaucipir in 2013, for the evaluation of cognitive impairment of patients being evaluated for AD. To date, the analytical method for measuring tau accumulation was static tau PET imaging based on SUVR approaches [34,35,36]. Recently, a Tau^IQ^ algorithm performed much better compared with traditional SUV ratio (SUVR) to detect tau changes and found the strongest correlation with MMSE and CDR-SB [37]. While the Tau^IQ^ algorithm is powered to detect a net reduction in tau accumulation in clinical trial settings, it is not optimized to diagnose global Tau changes in AD patients receiving Donanemab [37]. 

Amyloid-related imaging abnormalities (ARIA) have previously been reported in AD when treated with amyloid-β reduction immunotherapies. While the exact pathophysiology giving rise to ARIA is unclear, the APOE ε4 allele is a risk factor for cerebral amyloid angiopathy (CAA) and microhemorrhage found in postmortem human studies. ARIA-E refers to cerebral edema or sulcal effusion whereas ARIA-H refers to hemosiderin deposits as a result of hemorrhage within the brain parenchyma or pial surface. Both ARIA-E and ARIA-H may share underlying mechanisms of action which may be an increase in vascular permeability. Overall, ARIA may be related to amyloid clearance therapies and requires further elaboration to understand the long-term implications of such adverse events on the prognosis of patients with AD. In terms of the safety of Donanemab, phase I/II trials did not find any significant differences in serious drug-related adverse events between Donanemab and placebo arms including all-cause mortality. However, two types of drug-related adverse events, including any ARIA-E and ARIA-H, were present in 26.1–26.7% and 26.1–30.5% of the patients, respectively, across the phase I/II trials reported. As such, there is a prominent risk of ARIA when given amyloid reduction therapies which is consistent with alterations in vascular amyloid burden. 

### 4.1. Limitations and Recommendations 

There are a few limitations in this study. A majority of the patients in the phase II TRAILBLAZER-ALZ (NCT03367403) trial were APOE ε3 carriers (50.7%, N = 137). Typically, the most common allele frequency is APOE ε3 with an allele frequency of 67–87%. However, the APOE ε4 allele is higher in African Americans with a frequency of 40–65%. The APO ε4 allele is the most toxic of all allelic variants, ε2–4, and is associated with an increased amount of Aβ including a more toxic oligomeric form found in central nervous system (CNS) autopsies of AD patients posthumously. As such, the efficacy of Aβ reduction therapies can be differential across the major allelic variants, which has not been explored in current phase I/II trials of Donanemab. Post hoc analyses conducted by Shcherbinin et al. found a significant reduction in the maximum percentage decrease in amyloid plaque from baseline in APOE ε4 carriers (44%; 95% CI, 24–59%; *p* < 0.001). This post hoc model suggests beneficial efficacy based on different allelic variants which require further consideration in upcoming trials. Patients in the phase II TRAILBLAZER-ALZ (NCT03367403) trial were selected based on the flortaucipir PET screening criteria, which narrows down patients to specific clinical severity. Patients with the highest tau levels were not included, and this subset of patients may have different clinical responses. The sample did not have racial/ethnic diversity findings so cannot be generalized to all populations. The sample size across and within trials was small and the amyloid reduction benefits were likely too small to be of substantial effect due to a lack of statistical power. 

It is also of consideration that notable cognitive benefits may not be detected within the time frame of phase I/II trials included in this study as an attribute of amyloid reductions. If longer intervals are required for follow-up beyond that seen in this study at 76 weeks, longer trials may need to consider extending the duration in phase III settings. Furthermore, the patients included in phase I/II trials were not differentially categorized based on the duration and severity of AD. It is reasonable to expect early-stage AD patients to derive significant benefits from Donanemab in terms of greater cognitive benefit from amyloid level reductions due to a lower baseline amyloid burden. This may be due to the complete amyloid clearance in these populations as amyloid reduction followed a percentage reduction pattern such that lower baseline amyloid levels in patients were met with complete amyloid clearance. The consequent impact of complete amyloid reduction can greatly reduce global tau accumulation which can be tested in future clinical trials. Essentially, the interplay between amyloid-β (Aβ) plaques and global tau levels is of great interest when targeting amyloid-β (Aβ) reduction therapies such as Donanemab. 

### 4.2. Known Risk Factors of Donanemab

During the preliminary testing period, a multiple-dose study of 37 patients reported that six patients (16.2%) had infusion reactions with chills, dizziness, flushing, fever, and rash, with anti-drug antibodies in the plasma. As a drug class, Donanemab may lead to rare counts of cerebral microhemorrhage cases, vasogenic cerebral edema events, and also superficial siderosis events. These are reported on average at a rate of 10–15% in the included trials of this systematic review [23]. Other than these, the only other reported risks include amyloid-related imaging abnormalities, which are the most common treatment-emergent events.

### 4.3. Future Directions

There are currently 488 ongoing clinical trials aimed at improving the quality of life among patients with Alzheimer’s disease. An index of ongoing pharmacological clinical trials for Alzheimer’s disease is presented in Appendix A. With the plethora of ongoing trials, Donanemab adds one more therapy line to the mix. Slowing the progression of the disease has been a large unmet need of our time. While Alzheimer’s disease therapies have largely been controversial in their approval status by the FDA, the future needs more accessible and effective therapies. Anti-amyloid treatments, as mentioned in our systematic review, are one of the first phase III trials to be performed in pre-clinical Alzheimer’s disease. With upcoming trials and patient dissemination of novel therapeutics, it is essential that all clinical testing carries out considerable planning, longitudinal assessments, data storage, and management. The findings of ongoing work will help in revealing the success of Donanemab in a more population-based setting to improve the diversity and retention of the intervention, along with the rates of referrals to clinical trials.

## 5. Conclusions

This systematic review critically reviewed current evidence of Donanemab in 396 patients with varying degrees of Alzheimer’s disease (AD) across four studies in Phase I/II settings. A total of 228 patients received Donanemab whereas 168 of the patients were given a placebo. The gender and age ratio were well-balanced across the trials. The patients that were included had prodromal, mild, or moderate AD categorized by MMSE, FCSRT-IR, and CDR as well as amyloid plaques with or without tau pathology on ^18^F-flortaucipir PET scan. There was a favorable reduction in amyloid plaque levels which depended on baseline amyloid levels such that patients with lower amyloid plaque levels were found to have complete amyloid clearance. Other favorable outcomes were a reduction in the accumulation of overall tau levels and relatively lower functional/cognitive decline with Donanemab. Further phase III trials must explore the interplay between amyloid and tau levels as well as associated clinical outcomes to derive meaningful outcomes for AD patients. 

## Figures and Tables

**Figure 1 healthcare-11-00032-f001:**
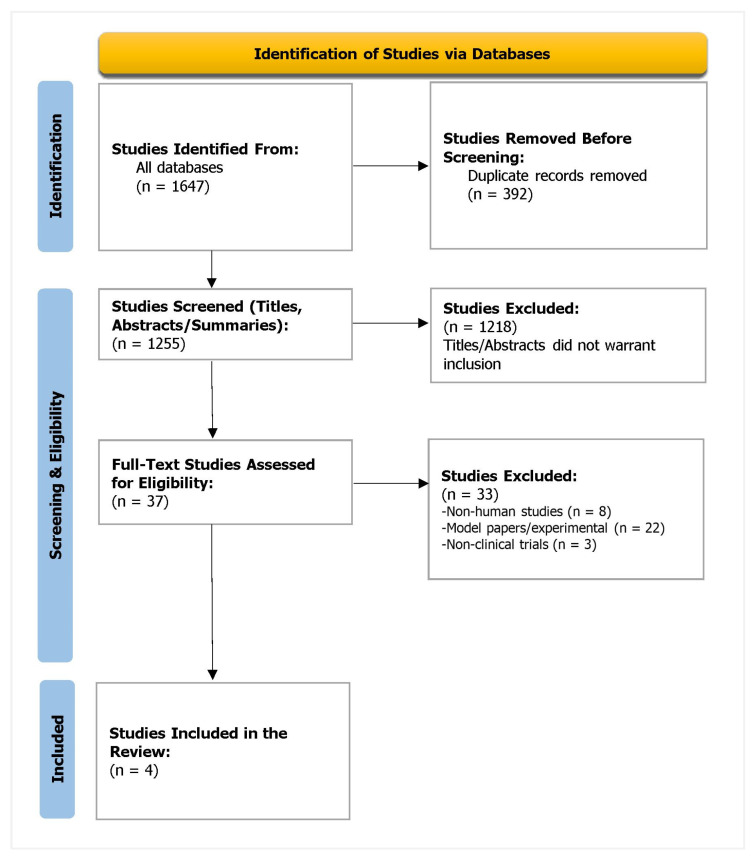
PRISMA flowchart depicting the study selection process.

**Figure 2 healthcare-11-00032-f002:**
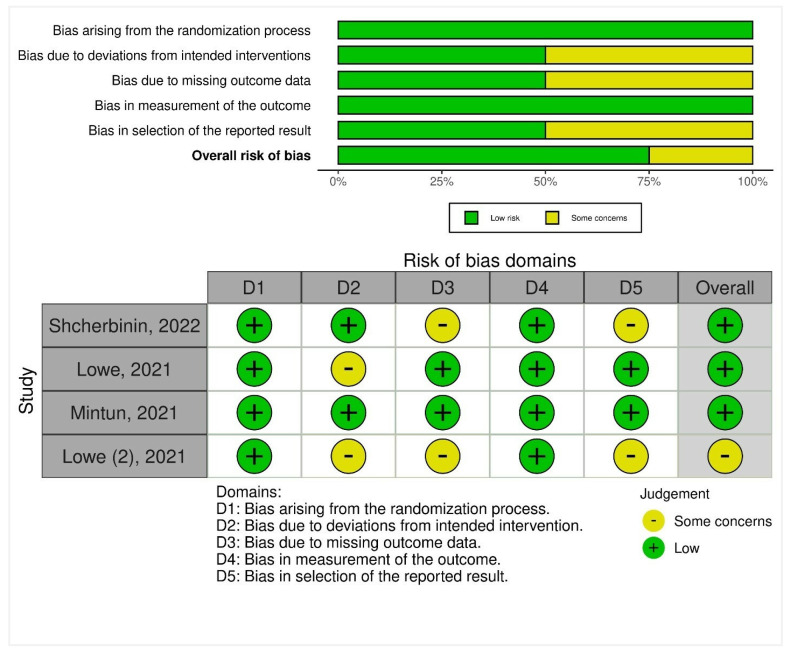
Risk-of-bias assessment of RCTs using the ROB-2 tool. Traffic light plot of study-by-study bias assessment. Weighted summary plot of the overall type of bias encountered in all studies.

**Table 1 healthcare-11-00032-t001:** Baseline characteristics and overview of the included trials.

Sr. No.	Author	Year	Title	Journal	Phase	Design	Inclusion Criteria	Pharmacologic Intervention	Outcome Measures	Follow-up
1	Lowe (1)	2021	Donanemab (LY3002813) dose-escalation study in Alzheimer’s disease	*Translational Research & Clinical Interventions*	Phase 1 study	Subject- and investigator-blind, randomized, placebo-controlled, parallel-group, a seven-arm study of single-dose, followed by a multiple-dose, dose-escalation study	Men or non-fertile women ≥50 years of age with evidence of memory impairment on FCSRT-IR picture version, MMSE score of 16 to 30, and a florbetapir PET scan consistent with amyloid pathology	Arm 1: 0.1 mg/kg IV (intervention: four patients, placebo: two patients); Arm 2: 0.3 mg/kg IV (intervention: seven patients, placebo: two patients) Arm 3: 1 mg/kg IV (intervention: nine patients, placebo: two patients) Arm 4: 3 mg/kg IV (intervention: 11 patients, placebo: 3 patients) Arm 5: 10 mg/kg IV (intervention: six patients, placebo: three patients) Arm 6: 3 mg/kg SC (intervention: eight patients) Arm 7: 1 mg/kg IV in six healthy volunteers	Brain amyloid plaque levels with Florbetapir-PET with SUVR; ADAS-Cog-14; MMSE; FCSRT-IR	48-week period
2	Lowe	2021	Donanemab (LY3002813) Phase 1b Study in Alzheimer’s Disease: Rapid and Sustained Reduction of Brain Amyloid Measured by Florbetapir F18 Imaging	*The Journal of Prevention of Alzheimer’s Disease*	Phase 1b	Three-part, patient- and investigator-blind, randomized within cohort, placebo-controlled, parallel-group, six-arm, single and multiple-dose study	Men or non-fertile women ≥50 years with evidence of memory impairment on FCSRT-IR, picture version, MMSE score of 16–30, CDR of 0.5–2, memory box score ≥0.5, and a florbetapir PET scan consistent with amyloid pathology	Cohorts 1–3 (single, IC dose of Donanemab 10 mg/kg, 20 mg/kg, 40 mg/kg) or placebo; Cohort 4 (multiple IV doses of Donanemab 10 mg/kg) or placebo every 2 weeks for 24 weeks; Cohorts 5–6 (multiple IV doses of Donanemab, 10 mg/kg, 20 mg/kg) or placebo every 4 weeks for 72 weeks	Brain amyloid plaque levels with ^18^F-flortaucipir PET scan with SUVR values; CDR; MMSE; FCSRT-IR; ADAS-Cog-14; ADCS-MCI-ADL-24; NTB	72 weeks (Cohorts 1 and 2); 24 weeks (Cohort 3); 48 weeks (Cohort 4); 12 weeks (Cohorts 5–6)
3	Mintun	2021	Donanemab in Early Alzheimer’s Disease	*The New England Journal of Medicine*	Phase 2 trial (NCT03367403)	Multicenter, randomized, double-blind, placebo-controlled trial	Patients 60 to 85 years of age who had early symptomatic AD, defined as prodromal AD or mild AD with dementia, and had an MMSE score of 20 to 28; flortaucipir PET scans with evidence of pathologic tau deposition but with quantitative tau levels between 1.10 and 1.46 except in advanced AD where tau levels ≤1.10 included with elevated amyloid levels (equivalent to ≥37 CL)	1:1 ratio to receive either Donanemab (700 mg for the first three doses and 1400 mg thereafter) or placebo, administered intravenously every 4 weeks for up to 72 weeks	Change from baseline to 76 weeks in iADRS score *; Change from baseline in CDR-SB scores; ADAS-Cog_13_, the ADCS-iADL, and the MMSE	76 weeks
4	Shcherbinin	2022	Association of Amyloid Reduction After Donanemab Treatment With Tau Pathology and Clinical Outcomes: The TRAILBLAZER-ALZ Randomized Clinical Trial	*JAMA Neurology*	Phase 2 (NCT03367403)	Multicenter, double-blind, phase 2, placebo-controlled, randomized clinical trial	Participants with AD who had an intermediate tau level (moderate AD patterns based on visual assessment and SUVR between 1.10 and 1.46, inclusive, or advanced AD patterns and SUVR ≤1.10) plus elevated amyloid level (equivalent to ≥37 CL)	Donanemab dosing was given every 4 weeks: 700 mg for the first 3 doses, then 1400 mg for up to 72 weeks.	Change from baseline in the score on the iADRS; Change in amyloid, tau, and clinical decline after Donanemab intervention	48-week period

* Integrated Alzheimer’s Disease Rating Scale (iADRS; range, 0 to 144, with lower scores indicating greater cognitive and functional impairment). Acronyms: AD: Alzheimer’s disease; ADAS-Cog_13_: the 13-item cognitive subscale of the Alzheimer’s Disease Assessment Scale; ADAS-Cog_14_: the 14-item cognitive subscale of the Alzheimer’s Disease Assessment Scale; ADCS-iADL: the Alzheimer’s Disease Cooperative Study–Instrumental Activities of Daily Living Inventory; ADCS-MCI-ADL-24: Alzheimer’s Disease Cooperative Study–Activities of Daily Living–Mild Cognitive Impairment 24-item version; CDR-SB: Clinical Dementia Rating Scale–Sum of Boxes; CL: Centiloids; FCSRT-IR: Free and Cued Selective Reminding Test Immediate Recall; MMSE: Mini-Mental State Examination; NTB: Neuropsychological Test Battery; PET: Positron emission tomography; SUVR: Standardized uptake value ratio.

**Table 2 healthcare-11-00032-t002:** Patient characteristics, and efficacy/safety outcomes of the trials.

Sr. No.	Author, Year	Sample Size	Age (Years)	Gender (Female)	Severity of AD	APOE-ε4 Carriers	Efficacy	Safety
1	Lowe (2), 2021	N = 63; Donanemab, n = 51; placebo, n = 12	69.7 ± 16.4 years	33 (52.4%)	MCI/Mild: 40/51 (78.4%) patients who received Donanemab and 9/12 (75%) patients who received placebo; moderate: 5/51 (9.8%) patients who received Donanemab and 3/12 (25%) who received placebo	NR	Donanemab was well-tolerated up to 10 mg/kg; Single-dose administration from 0.1 to 3.0 mg/kg yielded a mean terminal elimination half-life of ~ 4 days and this increased to ~ 10 days at 10 mg/kg; only 10-mg/kg dosage showed changes in amyloid PET, with a mean SUVR change of −0.26 (SD: −0.26) and mean CL change of −44.4 (SD: 14.2); around 90% of subjects developed anti-drug antibodies at 3 months after a single dose	No deaths or drug-related serious adverse events were reported; 6 of 37 patients (16.2%) who received IV Donanemab had mild-to-moderate infusion reactions; two patients (3.9%) in the intervention arms had asymptomatic ARIA-microhemorrhage; four participants (6.4%) across the entire cohort had serious adverse events not related to study drug, including hip fracture, cervical vertebral fracture, urinary tract infection, and noncardiac chest pain
2	Lowe, 2021	N = 61; Donanemab, n = 46; placebo, n = 15	73.2 ± 8.1 years	34 (55.7%)	Mean (SD) MMSE score: 21.1 (4.0)	47/61 (77.0%) patients; 11 homozygotes and 36 heterozygotes	Amyloid PET mean changes: at 24 weeks, from baseline for single doses were: −16.5 CL (SE = 11.22) with 10 mg/kg, −40.0 CL (SE = 11.23) with 20 mg/kg, and −49.6 CL (SE = 15.10) with 40 mg/kg; at 24 weeks, multiple dosage cohorts had −55.8 CL (SE = 9.51) with 10 mg/kg Q2w *, −50.2 CL (SE = 10.54) with 10 mg/kg Q4w, and −58.4 CL (SE = 9.66) with 20 mg/kg Q4w **; complete amyloid clearance (threshold of below 24.2 CL) established in 11 of 46 (23.9%) patients (one patient in the 20/mg single dose, one patient in the 40 mg/kg single dose, two patients in 10 mg/kg Q2w, two patients in 10 mg/kg Q4w, and five patients in 20 mg/kg Q4w); 45 out of 46 patients had positive TE-ADA with Donanemab	Seven serious adverse events in six patients (9.8%) were reported across the entire cohort (one patient died due to non-treatmen-related myocardial infarction, one had intermittent symptomatic cerebral edema (ARIA-E), and four patients had non-treatment-related events; 12 of 46 intervened patients (26.1%) developed vasogenic edema (ARIA-E) in all dosing regimens but the 10 mg/kg single-dose arm; 10 of 46 patients (21.7%) had microhemorrhage events (ARIA-H) across all dosing arms; 2 of 46 (4.4%) patients had superficial siderosis (ARIA-H) in the 10 and 20 mg/kg Q4w arms; 2 of 46 patients (4.4%) in the 20 mg/kg Q4w arm discontinued Donanemab due to TRAE
3	Mintun, 2021	N = 272; Donanemab, n = 131; placebo, n = 126	75.2 ± 5.5 years	145 (53.3%)	Mean (SD) MMSE score: 23.5 ± 3.1 (13.0–30.0)	197/270 (73.0%) patients; 141 heterozygotes and 56 homozygotes	Difference between the Donanemab and placebo groups in the change from baseline at 76 weeks for iADRS was 32% (*p* = 0.04) in favor of Donanemab, calculated through the following scores: a reduction of 6.86 from 106.2 baseline iADRS scores in the intervention group and 10.06 reduction from a baseline score of 105.9; difference between the Donanemab and placebo groups in change from baseline to 76 weeks were −0.36 for the CDR-SB score, −1.86 for the ADAS-Cog_13_ score, 1.21 for the ADCS-iADL score, and 0.64 for the MMSE score; there was an 84.13 CL reduction in the Donanemab group on Florbetapir PET from baseline scores of 108 CL; amyloid negative status (amyloid plaque level of < 24.10 CL) was found in 52 (40.0%) patients, 78 (59.8%) patients, and 89 (67.8%) patients at 24, 52, and 76 weeks; there was no change in global tau load from baseline to 76 weeks as assessed by flortaucipir PET; at 52 and 76 weeks, there was a greater decrease in whole-brain volume and greater increase in ventricular volume in the Donanemab group vs. the placebo group	No significant difference between the Donanemab group and the placebo group in the incidence of death or serious adverse events; 119 of 131 participants (90.8%) in the Donanemab group and 113 of 125 participants (90.4%) in the placebo group had at least one adverse event; 35 of 131 (26.7%) participants developed an ARIA-E in the Donanemab group of whom 8 (6.1%) participants were symptomatic and 1 of 125 (0.8%) participants developed an ARIA-E in the placebo group; 40 of 131 (30.5%) participants had an ARIA-H event in the Donanemab group and 9 of 125 (7.2%) participants had an ARIA-H event in the placebo group; cerebral microhemorrage was present in 10 of 131 participants (7.6%) and 3 of 125 participants (2.4%) in the Donanemab and placebo group, respectively; superficial siderosis was present in 18 of 131 (13.7%) participants in the Donanemab group and 4 of 125 (3.2%) participants in the placebo group; infusion-related reaction was present in 10 of 131 (7.6%) participants in the Donanemab group and did not occur in the placebo group
4	Shcherbinin, 2022	N = 272; Donanemab, n = 131; placebo, n = 126; combination, n = 15	75.2 ± 5.5 years	145 (53.3%)	Same as Mintun (2021)	Same as Mintun (2021)	46 of 115 (40%) of Donanemab participants reached a complete amyloid clearance threshold of 24.1 CL (r: –0.54), placebo-treated participants did have changed amyloid clearance change appreciably (r: –0.194); achieved amyloid clearance was sustained with a mean rate of reaccumulation of 0.02 CL (SD: 7.75) over a 1-year period; those who achieved an amyloid level ≤ 11 CL at week 24 would require a mean time of 3.9 years [95% CI: 1.9–8.3 years] to regain amyloid plaque levels above the threshold (>24.1 CL); a 34% slowing of overall tau level, measured using SUVR, was observed for Donanemab compared to placebo in the entire cohort at 76 weeks	All-cause mortality was lower in Donanemab-intervened participants (n = 1, 0.76%) compared to placebo (n = 2, 1.6%); no differences were reported in serious adverse effects among Donanemab (n = 26, 19.85%) and placebo (n = 25, 20%) groups

* Q2: Every 2 weeks. ** Q4: Every 4 weeks. Acronyms: AD: Alzheimer’s disease; ADAS-Cog_13_: the 13-item cognitive subscale of the Alzheimer’s Disease Assessment Scale; ADCS-iADL: the Alzheimer’s Disease Cooperative Study–Instrumental Activities of Daily Living Inventory; ARIA: Amyloid-related imaging abnormalities; ARIA-E: Amyloid-related abnormalities due to cerebral edema; ARIA-H: Amyloid-related imaging abnormalities due to hemosiderin deposition and hemorrhage; CDR-SB: Clinical Dementia Rating Scale–Sum of Boxes; CL: Centiloids; iADRS: Integrated Alzheimer’s Disease Rating Scale; MCI: Mild cognitive impairment; MMSE: Mini-mental state examination; PET: Positron emission tomography; SD: standard deviation; SE: Standard error; SUVR: Standardized uptake value ratio; TE-ADA: Treatment-emergent anti-Donanemab antibodies; TRAE: Treatment-related adverse events.

## Data Availability

Not applicable.

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
