# Peer review of "Donanemab for Alzheimer’s Disease: A Systematic Review of Clinical Trials"

_healthcare, 2022, doi:10.3390/healthcare11010032_

Round 1

Reviewer 1 Report

The authors comprehensively reviewed the clinical trials of Donanemab for Alzheimer's disease (AD). Four studies were included to study the effects of Donanemab on the delayed cognitive and functional decline in AD. The authors clearly discuss the important findings of the study and propose further recommendations for future studies. The adverse outcomes between these trials were briefly descripted. The authors clearly discuss the important findings of these studies and propose further recommendations for future studies. The limitations of this review were discussed clearly. The English expression of the article is fluent and clear.

There is a major comment need to addressed: the last paragraph of the introduction was not completed.

There are two minor points need to improved.

  1. The resolution of Figure 1 and Figure 2 is not good enough.
  2. The descreiption of Q2weekly and Q4weekly should be added in the footnotes of table 2.

In summary, this review present a different point of view regarding the effects of Donanemab in Phase I/II trials. 

Author Response

Reviewer 1 Comments and Author Responses

Comment 1: The authors comprehensively reviewed the clinical trials of Donanemab for Alzheimer's disease (AD). Four studies were included to study the effects of Donanemab on the delayed cognitive and functional decline in AD. The authors clearly discuss the important findings of the study and propose further recommendations for future studies. The adverse outcomes between these trials were briefly descripted. The authors clearly discuss the important findings of these studies and propose further recommendations for future studies. The limitations of this review were discussed clearly. The English expression of the article is fluent and clear.

Response 1: Thank you for your kind comments.

Comment 2: There is a major comment need to addressed: the last paragraph of the introduction was not completed.

Response 2: The last paragraph has been completed. One sentence got cut out in transition. You are free to look at the updated version highlighted in yellow. Thank you for your due diligence.

Comment 3: There are two minor points need to improved.

  1. The resolution of Figure 1 and Figure 2 is not good enough.
  2. The descreiption of Q2weekly and Q4weekly should be added in the footnotes of table 2.

Response 3: The resolution of both Figure 1 and 2 is now 1200 DPI, which is the maximum best resolution for any figure. I hope that clarifies your concern! :) 

The description for Q2 and Q4 has been added to the footnote of Table 2; the changes are highlighted in yellow.

Comment 4: In summary, this review present a different point of view regarding the effects of Donanemab in Phase I/II trials. 

Response 4: Thank you for your kind comments.

To the Esteemed reviewer, I truly thank you for taking out the time to review and improve our paper exponentially. The paper has been proofread (word to word) as well. I anticipate your timely response.

Kind Regards, 

Dr. Zouina Sarfraz

Reviewer 2 Report

In the review, the authors treated the latest evidence of Donanemab, a humanized antibody that

targets the reduction of Aβ plaques, in AD patients. The authors conducted bibliography research across PubMed/MEDLINE,  CINAHL  Plus,  Web of  Science,  Cochrane,  and  Scopus. They take into consideration 4 clinical trials for a total of 396 patients who received either Donanemab or a placebo. A general AD condition improvement was registered after Donanemab. The authors wanted to summarize the design and inclusion criteria, dosing regimens, primary and secondary outcome measures, and the efficacy of the drug.

Major revisions

The 2 citation is wrong reported in the references. It doesn’t contain authors or journals or pages.

Page 2 lines 73-74. What is the “umbrella methodology”? Which criteria were applied for the additional review of the reference list?

Minor revisions

Regarding the spelling of the name Donanemab, the authors should decide whether to write it in upper or lower case and maintain this decision throughout the text.

Page 2 line 65. The phrase is incomplete.

Page 2 lines 73-74, Page 2 lines 92, Page 3 lines 99. The first letter of the phrases must be changed with an uppercase character.

Page 15 line 232 there is a punctuation point extra

I suggest a general grammar and punctuation check.

The tables could be improved

Figure 2. the color legend is too small and blurry

Author Response

Reviewer 2 Comments and Author Responses:

In the review, the authors treated the latest evidence of Donanemab, a humanized antibody that targets the reduction of Aβ plaques, in AD patients. The authors conducted bibliography research across PubMed/MEDLINE,  CINAHL  Plus,  Web of  Science,  Cochrane,  and  Scopus. They take into consideration 4 clinical trials for a total of 396 patients who received either Donanemab or a placebo. A general AD condition improvement was registered after Donanemab. The authors wanted to summarize the design and inclusion criteria, dosing regimens, primary and secondary outcome measures, and the efficacy of the drug. Major revisions:

Comment 1: The 2 citation is wrong reported in the references. It doesn’t contain authors or journals or pages.

Response 1: The second citation has been updated. Thank you for noting the error. It is now good to go.

Comment 2: Page 2 lines 73-74. What is the “umbrella methodology”? Which criteria were applied for the additional review of the reference list?

Response 2: The term “umbrella methodology” has been removed; what we mean is - the reference lists were additionally reviewed. This has been both updated and corrected in the paper itself.

Comment 3: Minor revisions: Regarding the spelling of the name Donanemab, the authors should decide whether to write it in upper or lower case and maintain this decision throughout the text.

Response 3: Donanemab has been capitalized throughout the text. Feel free to have a look!

Comment 4: Page 2 line 65. The phrase is incomplete.

Response 4: The phrase has been completed. I thank you very much for your due diligence.

Comment 5: Page 2 lines 73-74, Page 2 lines 92, Page 3 lines 99. The first letter of the phrases must be changed with an uppercase character.

Response 5: Thank you for doing your due diligence; it has been updated.

Comment 6: Page 15 line 232 there is a punctuation point extra, I suggest a general grammar and punctuation check.

Response 6: It has been updated and my co-author who is an English native writer including myself (ENL certified) have both reread the entire draft to ensure no mistakes are left.

Comment 7: The tables could be improved

Response 7: I humbly disagree with this; the tables have been reviewed and reviewed for accurate presentation. I did however make changes in data and presentation to address your point! :)

Comment 8: Figure 2. the color legend is too small and blurry

Response 8: The figure has been updated to 1200 DPI, the highest resolution for an image we can produce. Please have a look at it now.

To the Esteemed reviewer, I truly thank you for taking out the time to review and improve our paper exponentially. The paper has been proofread (word to word) as well. I anticipate your timely response.

Kind Regards, 

Dr. Zouina Sarfraz

Reviewer 3 Report

The article on Donanemab for Alzheimer’s Disease: A Systematic Review of 2Clinical Trials is well designed and planned. The systematic review is well designed. The paper only needs changes in the result and discussion. Please add systematic analysis point wise in the result. In discussion add more potential and risk factors of drug and future directions.

Author Response

Reviewer 3 Comments and Author Responses: 

Comments: The article on Donanemab for Alzheimer’s Disease: A Systematic Review of Clinical Trials is well designed and planned. The systematic review is well designed. The paper only needs changes in the result and discussion. Please add systematic analysis point wise in the result. In discussion add more potential and risk factors of drug and future directions.

Responses: Please review the headings in the results: Everything is made in a systematic point-by-point format:

3.1. Design and Inclusion Criteria 

3.2. Dosing Regimens

3.3. Primary and Secondary Outcome Measures

3.4. Efficacy of Donanemab

3.5. Safety and Tolerability of Donanemab and

3.6. Risk of Bias Synthesis

I invite you to review sections 3-3.6.

--

Concerning the potential and risk factors of the drug and future directions, I have added two fresh new paragraphs highlighted in yellow. Please have a look. They are fully addressing the points you have raised and I have also added a Supplementary Table 1. That is meant to be an index of all ongoing clinical trials. I am confident that you will read the paper with interest.

To the Esteemed reviewer, I truly thank you for taking out the time to review and improve our paper exponentially. The paper has been proofread (word to word) as well. I anticipate your timely response.

Kind Regards, 

Dr. Zouina Sarfraz